# Interpretable Drug-to-Drug Network Features for Predicting Adverse Drug Reactions

**DOI:** 10.3390/healthcare11040610

**Published:** 2023-02-17

**Authors:** Fangyu Zhou, Shahadat Uddin

**Affiliations:** School of Project Management, Faculty of Engineering, The University of Sydney, Forest Lodge, NSW 2037, Australia

**Keywords:** drug-to-drug network, network centrality measures, adverse drug reactions, machine learning

## Abstract

Recent years have witnessed booming data on drugs and their associated adverse drug reactions (ADRs). It was reported that these ADRs have resulted in a high hospitalisation rate worldwide. Therefore, a tremendous amount of research has been carried out to predict ADRs in the early phases of drug development, with the goal of reducing possible future risks. The pre-clinical and clinical phases of drug research can be time-consuming and cost-ineffective, so academics are looking forward to more extensive data mining and machine learning methods to be applied in this field of study. In this paper, we try to construct a drug-to-drug network based on non-clinical data sources. The network presents underlying relationships between drug pairs according to their common ADRs. Then, multiple node-level and graph-level network features are extracted from this network, e.g., weighted degree centrality, weighted PageRanks, etc. After concatenating the network features to the original drug features, they were fed into seven machine learning models, e.g., logistic regression, random forest, support vector machine, etc., and were compared to the baseline, where there were no network-based features considered. These experiments indicate that all the tested machine-learning methods would benefit from adding these network features. Among all these models, logistic regression (LR) had the highest mean AUROC score (82.1%) across all ADRs tested. Weighted degree centrality and weighted PageRanks were identified to be the most critical network features in the LR classifier. These pieces of evidence strongly indicate that the network approach can be vital in future ADR prediction, and this network-based approach could also be applied to other health informatics datasets.

## 1. Introduction

The World Health Organization (WHO) defines adverse drug reactions (ADRs) as noxious and unintended responses that occur at doses normally used in the human body [1]. The long-term and costly process of medicine development is roughly divided into three phases: pre-clinical, clinical development, and post-marketing. It should be acknowledged that clinical trials and experiments are essential as they provide verified and trustworthy information through various ADR identification methods and analysis [2]. However, sample limitations and insufficient time at this stage still prevent some ADRs from being known [3,4]. This is also the reason ADRs are still reported to be one of the top 10 causes of fatality in the USA [5] and result in 2.7–15.7% of hospitalisation cases [6], even though a proper amount of time and money has been invested into the labs and all of these drugs have undergone substantial experiments and trials during early stages before they are successfully approved. Fortunately, ADR cases from either clinical trials or post-marketing surveillance (PMS) are recorded and easily accessed. Figure 1 illustrates the respective datasets in each stage that the general public and professional researchers have access to. Compared to the clinical tests, the post-marketing phase, on the other hand, provides more ADR information as the number of medicine takers grows, and ADR-affected people are able to report through electronic health records (EHRs), a spontaneous reporting system (SRS), or digital health platforms, or even share on a social media platform, such as Twitter. Despite the intrinsic flaws these databases may have, such as underreporting [7] and selective reporting [8], their significance should not be ignored, and they have played an important role in ADR mining in recent studies through machine learning and deep learning methods.

The recent literature has shown that all kinds of databases have been used, and it has been proven that all these datasets could function as enabling sources for predicting ADRs. Some popular datasets are regarded as benchmark datasets in this field of study, for example, SIDER presents indications and side effects [9], KEGG is a database of drugs’ metabolic pathways [10], etc. Some integrated databases collect information from various sources, such as Liu’s dataset [11] and Bio2RDF [12], etc. Some other researchers have also used EHR and SRS to predict ADRs. In terms of machine learning algorithms, random forest (RF), and support vector machine (SVM), in most cases, they have outperformed other techniques [11,13,14,15]. Deep learning algorithms have also shown remarkable abilities in predicting ADRs, especially convolutional neural networks (CNN) [16].

Another trend in recent years is that more efforts have been made to construct graphs based on various datasets of either clinical or non-clinical data. The emergence of graph-based machine learning methods characterized by complex topological structures, latent information in connections and multimodal features has presented fascinating opportunities to model a wider range of scenarios in different fields of study, as is also the case in the study of health informatics and adverse drug reactions research. Hu developed a signed-heterogeneous network and trained an encoder to learn drug embeddings and predict ADRs [17]. Kwak proposed a graph-based framework for ADR signal detection using healthcare claims data [18]. Zitnik constructed a drug-protein network that uses a link prediction method for modelling polypharmacy [19]. Zhou and Uddin demonstrated the importance of appropriately modelling network edges in order to achieve a better prediction outcome. In this regard, the incorporation of edge weights as a significant feature of edges was found to be crucial [20]. There are also many other existing network-based models in the literature, and most of them have tried to uncover drugs’ positions in a network via different methods to obtain better representations of the drugs and their underlying structural properties. In spite of these advances, little work has been done to deal with the latent relationships among drugs using a drug-to-drug network. Some network-based models might have strong power, but in many cases, it is their lack of interpretability that makes them hard to implement at an industry-standard level.

This paper presents an ADR prediction model using a weighted drug-to-drug network with the aim of utilising hidden information between drugs, which is accessed to determine if they can contribute substantially to the prediction of ADRs. First, we construct a bipartite graph from Liu’s dataset and then convert it into a drug-to-drug network, with the edges being the number of common ADRs for each associated drug pair. After that, the network features are concatenated to the original drug features in Liu’s dataset, thus forming new sets of features for learning. A comparison between experiments with network features and without network features is made, and this addresses the importance of the network features. In addition, we further explore which network measures are critical in achieving a better prediction result. This study’s findings will redound to the benefits of various healthcare stakeholders, including health informatics investigators, drug development researchers, health insurance organisations, and a broader community. The major contributions of this study are as follows: (1) First, this study is among the very first to construct weighted networks to model drugs’ relationships from integrated non-clinical datasets. (2) Second, the extraction of network features enables interpretable models since the importance of each network feature can be analysed when predicting a single drug’s ADRs. (3) Third, the edge weights were verified to be substantial in making predictions about a single drug’s ADRs, and they are also prominent in modelling the relationships between drug pairs.

## 2. Materials and Methods

An overview of this study is displayed in Figure 2. A bipartite graph was first constructed using Liu’s dataset [11], which was then projected on drugs, thus forming a weighted drug-to-drug network to be used in the proposed models. A detailed description of this process is included in Section 2.4. A total of ten network features were extracted, e.g., weighted degree centrality, eigenvector centrality, weighted PageRanks, etc. These network features were then concatenated to the drug features (the baseline data). Different combinations of the network features were then assessed using seven machine learning algorithms, including support vector machine (SVM), logistic regression (LR), etc. AUROC was used as the metric to determine which methods would yield the best results, based on which the features’ importance could be analysed as well.

### 2.1. Dataset Description and Pre-Processing

Liu’s dataset [11] contains 832 kinds of drugs, and each drug consists of six categories of features: chemical, enzyme, indication, pathway, target, and transporter. Their sources and statistics are listed in Table 1.

For all the features concatenated together, there were 2892 (881 + 111 + 869 + 173 + 786 + 72) dimensions to be considered for each drug and 1385 ADRs in this dataset, so one-hot vector features were created for each drug and their corresponding ADRs. This means that each drug is represented by a vector with a length of 2892 consisting of 1 or 0, with 1 indicating that this drug is associated with a particular enzyme, indication, etc., and 0 meaning they are not associated. Similarly, the ADRs of each drug are represented by a vector with a length of 1385 consisting of 1 or 0, with 1 meaning it is evident that this drug results in an identified ADR and 0 meaning otherwise.

### 2.2. PCA Applied to One-Hot Vectors of Drugs’ Features

As there were 2892 dimensions of the drug dataset, which was considered too many for the sake of the learning time and more prone to overfitting issues, principal component analysis (PCA) was applied to this dataset and then the feature data was converted into 10 dimensions that could help explain approximately 50% of the variance of the drugs’ features. Figure 3 illustrates the number of components chosen and the variance in the dataset that could be explained accordingly. PCA dramatically quickens the learning process while still accurately representing the characteristics of the original dataset. Comparing 10 components and 85 components (which could explain 80% of all the variances), it turned out the latter slightly improved the results with random forest but consumed much more time than the former. Therefore, 10 components were used for the rest of the experiments.

However, there are downsides to applying PCA to a one-hot vector, as the features after PCA made the learning models less interpretable because they did not contain the original forms of the features, thus making it impossible to analyse the defining drug features.

### 2.3. SMOTE for Imbalanced Data

This dataset was severely imbalanced, evidenced by the fact that some of the ADRs resulted from a couple of drugs only, and it was not guaranteed that test sets would always have a fair number of positive cases so that the machine learning models would be able to work properly.

Figure 4 shows the distribution of ADR frequencies. A significant number of ADRs appeared fewer than 15 times across the entire dataset, which is fewer than the expected number of positive samples for good machine learning models, as they could not be used for 10-fold cross-validation. Therefore, experiments with ADRs appearing more than 5 times against 5-fold cross-validation were conducted, and their results were compared with those occurring more than 15 times. Generally, the former slightly increased the accuracy and AUROC of the models. However, they did not reflect the actual usability of such models because the imbalanced nature tended to predict more to be negative, so they still had high accuracy.

Therefore, we did not experiment with the ADRs with frequencies of fewer than 15 times; accordingly, there were 519 ADRs predicted in the next experiments. However, the selected ADRs still suffered from the imbalanced issue, considering that 15 positive cases is much lower than a fair number compared to the total number of drugs, which was 832. Therefore, the synthetic minority oversampling technique (SMOTE) was employed to address this issue [21]. The way that SMOTE works is to select data points that are close in the feature space, draw a line between the selected points in the feature space, and then create a new sample at a point along that line, which is what synthetic means in this method. Specifically, a random example from the minority class was first chosen. Then the k nearest neighbours for that example were found (k = 5 in the current setting). A randomly selected neighbour was chosen, and a synthetic example was created at a randomly selected point between the two in the feature space.

Figure 5 is an example of the ADR at index 8 from Liu’s dataset. The first picture shows its original drugs’ data points, where there are only 61 positive cases but 771 negative ones. This gave rise to the problem of tending to predict false negatives for all positive cases while maintaining high accuracy and AUROC scores. After applying the SMOTE, the second picture shows how the data look with an oversampling ratio of 0.43, which was tested to be suitable for the next steps. This means that 270 (771 × 0.43 − 1) positive cases were created following the SMOTE algorithm. It should be noted that this method was only applied to the training data and was not used in the test data. It can also be observed that the synthesised data also show a pattern, which could be helpful for intuitive prediction.

### 2.4. Drugs Network Construction

A simplified version of the drug-to-drug network was constructed as shown in Figure 6, and the details are explained in the below sub-sections.

#### 2.4.1. Bipartite Graph Construction

A bipartite graph is a graph where the vertices can be decomposed into two sets such that no two vertices within the same set are connected. In our experiment, the two kinds of vertices were drugs and ADRs. The edges between the two sets denote the relationship between a particular drug and a specific ADR. Figure 6a shows a simplified example of 4 drugs and 4 related ADRs.

In this simplified instance, drug 1 is associated with ADR 1 only, meaning that it is evident that drug 1 will result in ADR 1 (sleepiness), but there is no sufficient evidence to prove that it is associated with ADRs 2 to 4. Similarly, drug 4 is associated with ADRs 2, 3, and 4 but not 1. Similar statements could also be made for other drugs, such as 2 and 3. However, as there is no information as to how reliable each association between the drugs and ADRs in this dataset is, we did not have the opportunity to model the edges as weighted, which is the reason the edges in the bipartite graph have the same thickness, meaning they have the same weight of 1.

#### 2.4.2. Bipartite Graph Projection

The next step was to project the bipartite graph onto ADRs, which formed a graph with only one set of nodes, which are drugs in this case. Figure 6b is an illustration of the network consisting of only drug nodes from the previous simplified example in Figure 6a.

When projecting onto the ADRs, we associated drugs that could essentially result in the same ADRs. The logic behind this is that the same ADR can be the harmful result of the same components or chemicals within the connected drugs. The thickness of the edges in this graph reflects how strongly related the corresponding drugs are. In terms of the simplified graph, drug 1 and drug 2 can cause ADR 1, so they are connected via an edge. Similarly, the same edges were created for these pairs: (drug 1, drug 3), (drug 2, drug 3), (drug 2, drug 4), and (drug 3, drug 4). However, pair (drug 3, drug 4) is denoted with a thicker edge, meaning their association is more substantial compared to other couples. The reason is that drug 3 and drug 4 could both result in ADR 3 and ADR 4, while in contrast, the other pairs have only one ADR in common. Finally, a visualization of the entire network is displayed in Figure 7, which was completed with Gephi [22]. The nodes in this figure represent different drugs, and the edges represent the links between them. The nodes are coloured differently to represent their clusters based on the modularity identified by the Louvain method [23], and higher modularity means the connections between nodes from the same cluster are denser and vice versa. In this graph, it can be seen that the drug nodes are sparsely scattered and not forming massive groups. The edges are coloured with different shades of orange, and the darker the edge is, the heavier weight this particular edge has, representing a stronger relationship between the nodes.

The problem that arises when abundant large weights exist in this graph is that the network features could be dominant regardless of the importance of the drug features, which makes the models tested less robust. Therefore, to assess the weights on a reasonable basis compared to other drug features, the edge weights were normalized based on the following calculation formula:$$\mathrm{Xnew}=\frac{X\;-\;\mathrm{Xmin}}{\mathrm{Xmax}\;-\;\mathrm{Xmin}}$$
where X is the original list of values and Xnew is the list of normalised values. Xmin refers to the smallest value in the original list, while Xmax refers to the largest. The resulting Xnew will have values between 0 and 1, with the smallest value changing to 0 and the largest to 1.

### 2.5. Network Features

The network feature were extracted and concatenated to the dataset. The addition of the 10 network features made the total number of features 20 in this dataset. The ten network features were degree centrality, weighted degree centrality, Eigenvector centrality, closeness centrality, clustering coefficient, betweenness centrality, hub, authority, triangles, and weighted PageRanks. Detailed calculations of these measures based on some prominent papers [24,25,26,27,28,29,30] can be found in the Appendix A.

### 2.6. Machine Learning Pipeline and Implementations

The experiment can be explained with the following pseudocode in Algorithm 1:
**Algorithm 1:** Pseudocode of the experiment designBeginFor each ADR doCount how many drugs are associated with it and save it in variable NIf N < 15 do Move to the next oneEnd ifIf N >= 15 do If positive cases/negative cases < 0.4 do  Apply SMOTE End if Apply learning algorithm and 10-fold cross-validationEnd if Calculate the average metrics for each algorithmEnd

Seven classical machine learning models were used (logistic regression, decision tree, XGBoost tree, random forest, support vector machine, k nearest neighbours, and artificial neural network). To implement these machine learning methods, we used python’s Scikit-learn library for the decision tree, XGBoost tree, random forest, support vector machine, and k nearest neighbours models and Keras for the ANN model.

The decision tree model bases its prediction on a tree-like process, where each branch uses one variable that determines which variable should be used next [31] until it decides which category this particular instance should fall into. In this experiment, the default hyperparameters in Scikit-learn were used as they were determined to outperform other alternatives. With the tree algorithm in place, random forest constructs a bunch of trees while keeping the variations under control [32]. Our experiment set the maximum depth of trees at 10, which was determined to be the best depth in this case. XGBoost is a scalable machine learning algorithm based on the decision tree model as well. It is a typical gradient boost algorithm implementation that helps approximate learning [33]. We used hyperparameter tuning to find the best performance for this machine learning method. Support vector machine finds the hyperplane in the multi-dimensions the data fall into and maximises the distances between the nearest data and the hyperplane, which helps separate the data into the desired categories [34]. Compared to these algorithms, k nearest neighbours is the most straightforward yet effective method, and it makes predictions based on the k nearest data points around the instance that is to be predicted [35]. In our experiment, the k was tuned to be 5. Artificial neural network employs a fully connected network that passes information through multiple layers, where the first layer of nodes corresponds to the input data, and the last layer is the prediction. It was proven that adding hidden layers efficiently increased the prediction accuracy by learning non-linear relationships and complex relationships [36]. When implementing this method, we used a three-layered network with one hidden layer; the optimiser used was Adam optimiser in Keras.

## 3. Results

As mentioned earlier, the experiments were performed using 10-fold cross-validation, which required each ADR to be associated with at least 15 drugs so that each test set could have at least one positive instance in most cases. The area under the receiver operating characteristic curve (AUROC) was used to measure the classifiers’ performance. The higher the AUROC score, the greater the ability the algorithm has to distinguish between positive and negative cases [37].

### 3.1. Network Overview

As per Section 2.4, a weighted network was constructed, with drugs represented by nodes and their relationships by undirected weighted edges. Each edge corresponds to the possibility that the connected drugs result in the same ADRs, and the weight indicates how many ADRs they both have. The thicker the edge, the more ADRs the related drugs have in common. The rationale is that both drugs might have effects on the same proteins with the same pathways, which is why they result in the same ADR, and hence why the drugs are connected. The statistics of the network can be found in Table 2.

These statistics show that this is a strongly connected network consisting of 832 nodes, between which there are 332,806 edges, making the average degree extremely high. This conclusion is also evidenced by the relatively small average path length, relatively high graph density, and high average clustering coefficient. The average path length means that, on average, from one node to another, it only takes 1.04 edges to arrive. The graph density shows the proportion of existing edges to all the possible edges that could make this graph fully connected. Therefore, 96% of the edges are present in this graph, making it nearly fully connected, which further indicates that the edge weights can provide more information rather than the existence of these edges.

### 3.2. Performance of the Proposed Methods Compared to the Baseline

#### 3.2.1. Network Features Assessment

The first experiment was to examine if adding network features would improve the AUROC score. The data used first were drug features only as the baseline. Adding each of the 10 network features created a new set of data for learning. The last set of features included all network features accordingly. The seven models were applied and tested for each of the 519 kinds of ADRs, and their results are shown in Table 3.

In this experiment, it was easily observed that the network features provided more insights than only using the drug features, regardless of which algorithms were used. Logistic regression achieved the highest AUROC when using the network features, which was 0.821, followed by SVM (0.819) and ANN (0.818). All of them were over 0.81, which is regarded as a good performance. The most significant increase was also achieved by logistic regression, which was an improvement from 0.595 to 0.821 by 0.226 in the AUROC score. Therefore, it is a safe conclusion that the proposed network features play a significant role in predicting ADRs. By scanning the results from the top to the bottom, we observed that a giant leap was achieved when weighted degree centrality was added to the dataset, which might mean that adding this particular feature is significant in ADR prediction. However, this does not mean other features are less effective. To obtain an insight into the logistic regression model, a detailed analysis of the feature selection was performed and is described in Section 3.2.3, and the features that are essential in the logistic regression model are discussed.

#### 3.2.2. ADRs Prediction Ranks from Logistic Regression Model

In the first experiment above, logistic regression showed an outstanding ability to predict ADRs compared to the other algorithms. The second experiment was to only use logistic regression and list the ADRs with AUROC scores greater than 90% when using network features. Table 4 details the top 10 ADRs identified with the highest AUROC scores when predicted with network features, together with their respective performance when without network features. In addition, the prediction accuracy rates are also attached.

This table shows that the ADR known as burning sensation (indexed as ADR id 961 in this dataset) was predicted with a considerable increase from 49% to 99.8%. Pyelonephritis (ADR id 753) also increased from 49.4% to 94.3%, and otitis externa (ADR id 645) increased from 50.6% to 93.9%. Even those predicted with a high AUROC score, such as miliaria (ADR id 564), also had a slight increase from 93.6% to 95.8% when network features were added.

What is also noticeable is that the accuracy from all these examples was comparatively high, with the highest being burning sensation, whose accuracy was 99%, followed by miliaria, which was predicted with an accuracy of 95.9%, and choreoathetosis (ADR id 951), which had an accuracy rate of 95.1%.

#### 3.2.3. Feature Importance

To verify if all the network features were important in the LR model, Table 5 details the mean *p*-value of each feature when logistic regression was used, which also helps to interpret the model in a detailed manner.

The *p*-value is a statistical measure determining whether a null hypothesis should be rejected. A small *p*-value (usually smaller than 0.05 or 0.01 in a more confident scenario) means that the hypothesis that this feature is not important should be rejected, which further indicates that this feature should be included in the feature selection. Based on this theory, the weighted degree centrality was the most important across all kinds of ADRs, closely followed by weighted PageRanks. The standard deviation also suggests that the weighted degree and weighted PageRanks are more closely scattered than the other features. To see if the distribution of these features’ *p*-values affected this conclusion, Figure 8 was created to show their respective boxplots. Based on this figure, it can be confidently stated that the weighted degree centrality and weighted PageRanks were both much lower than the other measures. These two were smaller because they were the only network features that took the edge weights into consideration and not only the existence of the edges, which in turn, provided more valuable information on the latent relationships between drugs. As can be seen in its calculation formula, the higher the weighted degree centrality, the more likely the two connected drugs have something in common, which finally results in a higher prediction rate.

## 4. Discussion

This study proposes a new network in which nodes represent drugs and edges represent their relationships and ten network features are utilised to predict the drugs’ related ADRs by integrating these features into the drug features’ dataset. After applying seven machine learning models to the newly created dataset, we calculated the mean AUROC score and found that by adding the network features, the mean AUROC metric increased regardless of which algorithm was applied. By further analysis of the network features’ p-values of the most promising algorithm, which was logistic regression in this instance, we found that the weighted degree centrality and weighted PageRanks served as the most critical factors that enabled better ADR predictions.

To the best of our knowledge, this is the first study to develop a drug-to-drug network based on their common ADRs and to extract the network features for machine learning. Some other networks have also been developed and analysed recently, either by using a large corpus to train a Word2vec model or using a large database for network construction. This study is different from others in that the network features were used, and it was found that these features could be important in the seven machine learning algorithms used, not to mention the weighted degree centrality and weighted PageRanks, which are considered more important than the drug features themselves. This graph-based method provided more interpretability to the logistic regression model.

When looking into the calculation of the ten network measures, we easily found that the weighted degree centrality and weighted PageRanks considered the edge weights, while other measures only considered whether there was a link or not, which is possibly the reason the p-values of these two measures were extremely low, meaning they were extremely important and could not be excluded when determining a good model for most of the ADR predictions. There could be biological reasoning for this in that the edge weights directly represent how closely connected the drugs are; the more closely related, the more likely they could result in the same ADRs because there is a greater chance they are working on the same proteins via the same pathways. Therefore, the edges, especially their weights, already contain much information about the drugs themselves.

It is also interesting that the correlation heat map (Figure 9) shows a high correlation between the weighted degree centrality and the weighted PageRanks. To be specific, their correlation coefficient was as high as 0.99994566. Regarding this as an undirected weighted network, the edges were treated as bidirectional to calculate the PageRanks. Therefore, intuitively, the PageRanks are more dependent on the edge weights if this score is considered as the possibility of an endpoint of a random walk.

It should be noted that this is not the only way to construct a network of drugs. Based on other properties, it is possible to build other networks, such as the common proteins the drugs work on. By considering the hierarchical structures of proteins, this network might prove to be more informative than this current one discussed in this paper, thus probably resulting in a higher increase in the ADRs’ prediction or at least offering an alternative approach for comparison studies. With the same thinking in mind, trying other databases for similar network construction is worthwhile. As other datasets might provide more drugs, a more comprehensive network could be built for further analysis and comparisons.

This study also has its limitations. PCA was applied to all the drugs’ features regardless of the feature categories, making it less interpretable and impossible to determine which drugs’ features were important for the ADR prediction. A possible solution to this problem is to separate the drug features into groups or categories. PCA can be applied to each group so that while quickening the training speed, it could also be identified which parts of the drugs’ features are important. Another limitation is that this study did not take the classifications and divisions of ADRs [38] into consideration when modelling the network. Although we have predicted particular ADRs using the drug information only, it can also be beneficial to use the information from their categories, which should be the direction of future research.

## 5. Conclusions

This study presents a new way of predicting ADRs using a network-based approach and integrating the network-based features into the drugs’ original features. We first constructed a bipartite graph that represents drugs and ADRs as two sets of nodes and their effect associations as edges and then projected the network onto ADRs so that a new weighted network was created representing drugs as nodes and their relationships as edges. After extracting the network information, such as the weighted degree centrality, etc., we applied seven machine learning algorithms, and the experiments found logistic regression to be the most promising in identifying ADRs based on the new datasets, and the weighted degree centrality and weighted PageRanks were proven to be the most important network features in predicting ADRs. The unique part of this approach is its high interpretability with the hand-crafted features and easy implementation in practice. In light of the potential of network measures in the scenario of a single drug’s ADRs predictions, extra research could be conducted to further showcase their effectiveness in predicting the ADRs that result from multiple drugs when the drugs are taken together. Some work is also needed to fully unveil the implication of the network measures, for example, using network features would be helpful to model the similarities between drugs.

## Figures and Tables

**Figure 1 healthcare-11-00610-f001:**
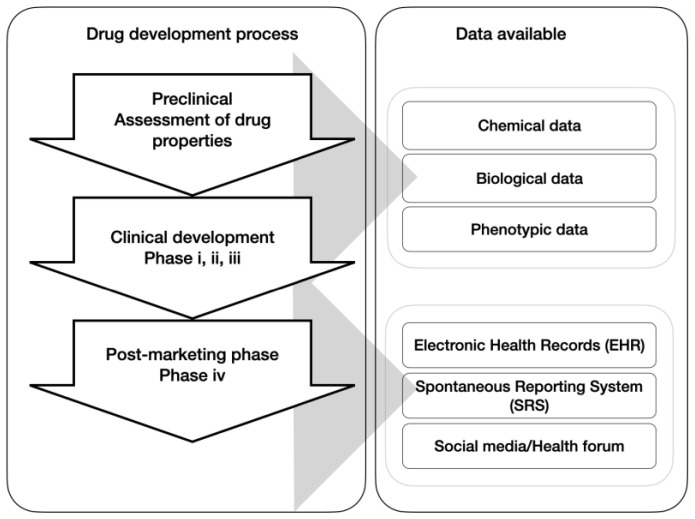
Drug development stages and the corresponding data generated in each stage.

**Figure 2 healthcare-11-00610-f002:**
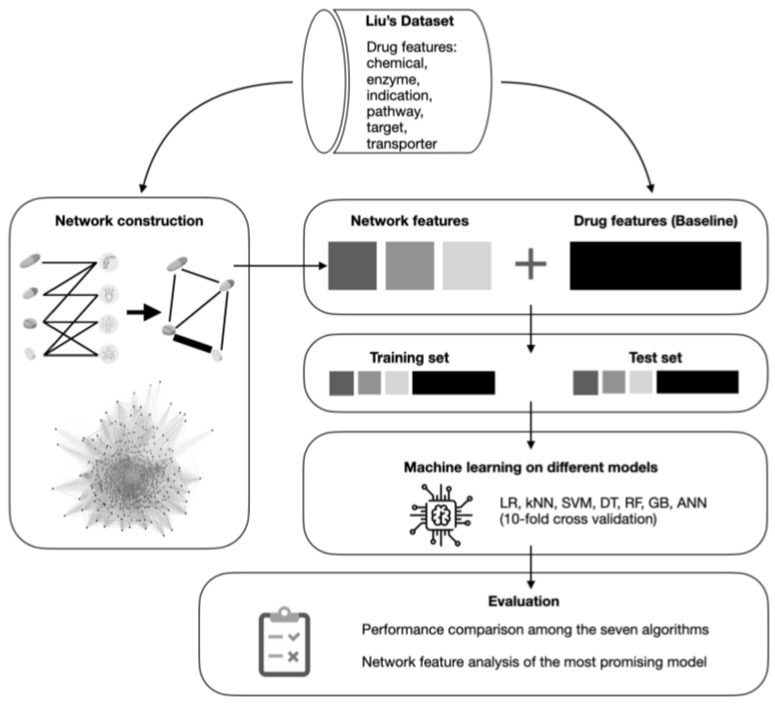
Overview of this study.

**Figure 3 healthcare-11-00610-f003:**
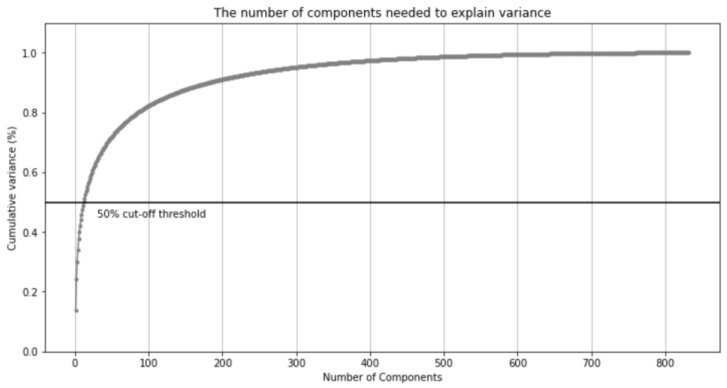
The number of components needed to explain variance. The threshold for the experiments was to explain 50% of the variance, which helped to quicken the training time.

**Figure 4 healthcare-11-00610-f004:**
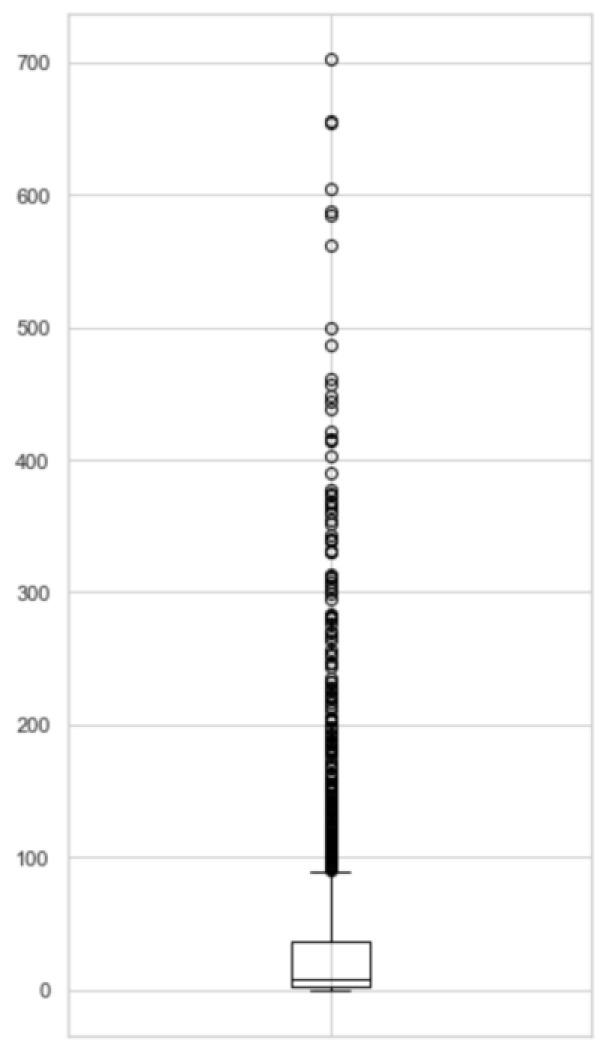
A boxplot showing the distribution of ADR frequencies. Each of the dots represents a kind of ADR, and the corresponding number on the vertical axis represents its frequency.

**Figure 5 healthcare-11-00610-f005:**
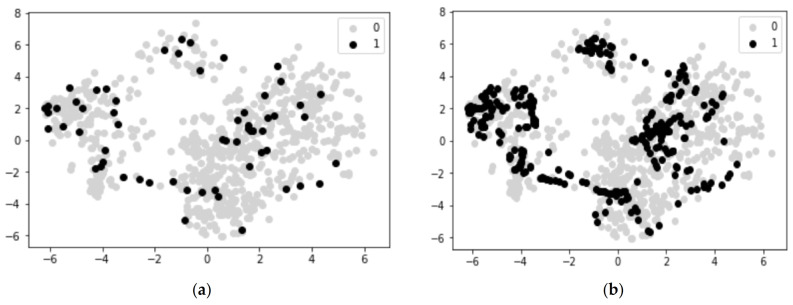
A comparison of the drugs’ data points of ADR 8 before and after applying SMOTE. (**a**) distribution of drugs associated with ADR 8 before SMOTE; (**b**) distribution of drugs associated with ADR 8 after SMOTE.

**Figure 6 healthcare-11-00610-f006:**
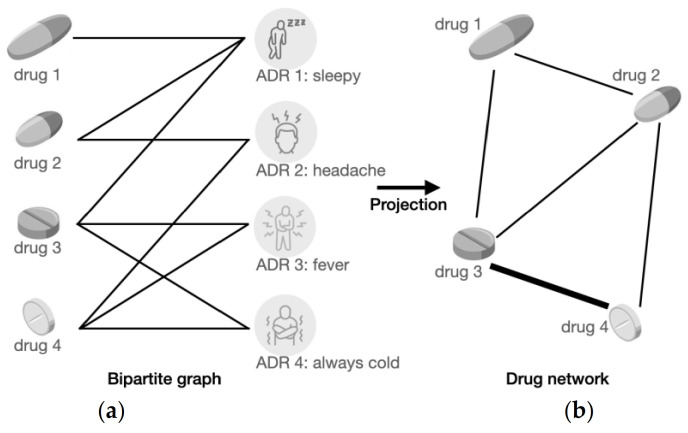
A simplified example of 4 drugs and their associated ADRs. (**a**) Simplified drug–ADR network; (**b**) simplified drug-to-drug network.

**Figure 7 healthcare-11-00610-f007:**
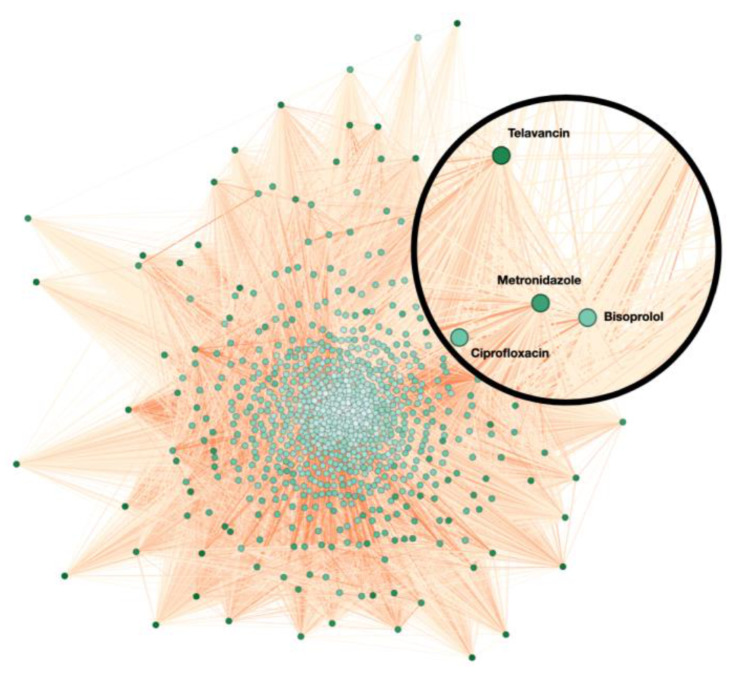
Visualisation of the drug-to-drug network.

**Figure 8 healthcare-11-00610-f008:**
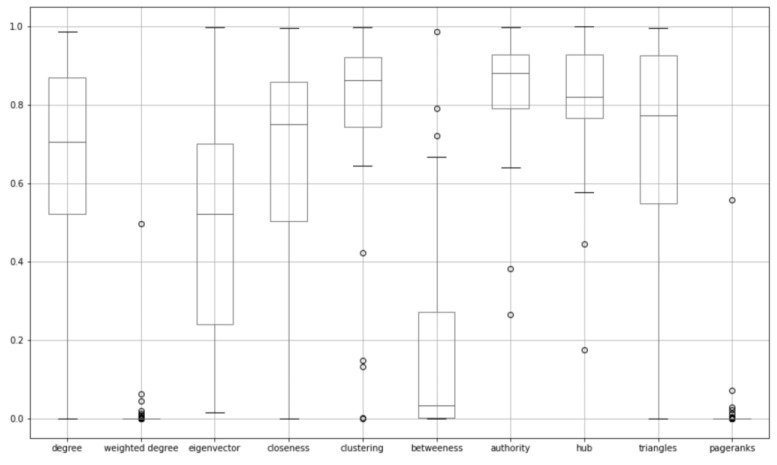
Boxplot of p-values of the 10 network features when logistic regression is applied to each of the ADRs.

**Figure 9 healthcare-11-00610-f009:**
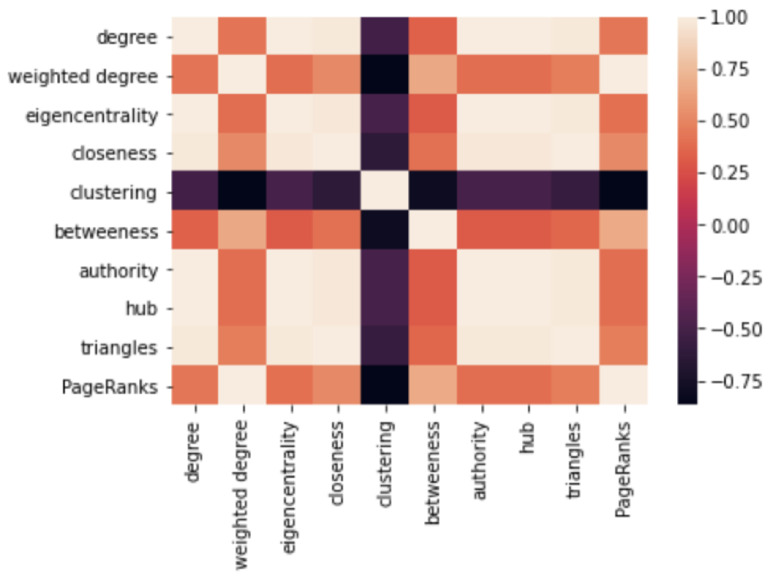
Correlation between the 10 network features.

**Table 1 healthcare-11-00610-t001:** Dataset sources and their respective number of features.

Feature Types	Sources	Number of Features
Chemical	KEGG	811
Enzyme	GeneBank	111
Indication	SIDER	869
Pathway	KEGG	173
Target	GeneBank	786
Transporter	GeneBank	72

**Table 2 healthcare-11-00610-t002:** Network properties’ statistics.

Properties of the Network	Statistics
Node count	832
Edge count	332,806
Average degree	800
Network diameter	2
Average path length	1.04
Graph density	0.96
Average clustering coefficient	0.97

**Table 3 healthcare-11-00610-t003:** Performance of the 7 classifiers tested on different combinations of datasets.

Datasets	Random Forest	Logistic Regression	SVM	kNN	Decision Tree	Gradient Boost Tree	ANN
Baseline	0.618	0.595	0.608	0.593	0.542	0.595	0.670
Baseline + D *	0.770	0.633	0.644	0.599	0.603	0.761	0.688
Baseline + DW *	0.802	0.814	0.811	0.727	0.623	0.787	0.813
Baseline + DWE *	0.803	0.815	0.813	0.727	0.621	0.786	0.813
Baseline + DWEO *	0.803	0.815	0.813	0.727	0.621	0.787	0.817
Baseline + DWEOC *	0.805	0.819	0.816	0.735	0.622	0.790	0.816
Baseline + DWEOCB *	0.805	0.819	0.816	0.737	**0.623**	**0.792**	0.816
Baseline + DWEOCBA *	0.805	0.819	0.816	0.737	**0.623**	**0.792**	0.817
Baseline + DWEOCBAH *	0.804	0.819	0.816	0.736	**0.623**	**0.792**	0.817
Baseline + DWEOCBAH T *	0.804	0.819	0.817	0.737	0.622	**0.792**	0.817
Baseline + DWEOCBAHTP *	**0.806**	**0.821**	**0.819**	**0.743**	0.622	0.791	**0.818**

* In the datasets column, D represents degree centrality, W represents weighted degree centrality, E represents eigenvector centrality, O represents closeness centrality, C represents clustering coefficient, B represents betweenness centrality, A represents authority, H represents hub, T represents triangle, and P represents weighted PageRanks.

**Table 4 healthcare-11-00610-t004:** Performance of logistic regression applied to the top 10 ADRs.

ADR id	ADR Name	AUROC without Network Features	AUROC with Network Features	Improvement	Accuracy with Network Features
961 [C0085624]	Burning sensation	0.490	0.998	0.508	0.990
951 [C0085583]	Choreoathetosis	0.740	0.966	0.226	0.951
564 [C0026113]	Miliaria	0.936	0.958	0.022	0.959
371 [C0018991]	Hemiplegia	0.568	0.947	0.379	0.919
1078 [C0162316]	Iron deficiency anemia	0.646	0.947	0.301	0.927
753 [C0034186]	Pyelonephritis	0.494	0.943	0.449	0.918
226 [C0012813]	Diverticulitis	0.813	0.941	0.128	0.929
645 [C0029878]	Otitis externa	0.506	0.939	0.433	0.922
204 [C0011334]	Tooth caries	0.519	0.937	0.418	0.921
541 [C0024894]	Mastitis	0.508	0.932	0.424	0.916

**Table 5 healthcare-11-00610-t005:** Network features’ mean p-values across all the ADRs when logistic regression was applied.

Network Features	Mean of *p*-Values	Standard Deviation of *p*-Values
Degree	0.673447	0.252389
Weighted degree	0.010291	0.063076
Eigenvector	0.511627	0.287661
Closeness	0.654633	0.274280
Clustering coefficient	0.793905	0.219335
Betweenness	0.190212	0.260536
Authority	0.849929	0.132538
Hub	0.823477	0.140945
Triangles	0.711467	0.264627
Weighted PageRanks	0.011244	0.070466

## Data Availability

Not applicable.

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
