# Peer review of "Interpretable Drug-to-Drug Network Features for Predicting Adverse Drug Reactions"

_healthcare, 2023, doi:10.3390/healthcare11040610_

Round 1
Reviewer 1 Report
This study attempted to design a new computational method to predict drug adverse drug reactions (ADRs). Authors employed ten network features derived from a drug-to-drug network. The network was constructed based on ADRs of all investigated drugs. The network features were combined with drug features (used in one previous), which were fed into seven classification algorithms to built the models. The ten-fold cross-validation results indicated that the employment of network features can improve the performance. Two network features were deemed to be more important than others. The manuscript is well-written and easy to understand. However, there are several issues.
1. In this version, authors used lots of “I”. As there are two authors for this manuscript, such expression is not proper. And it is better to avoid using “I” and “We”.
2. In Figure 3, the X-axis missed.
3. What is the meaning of “drug 8” in Figure 5? This figure shows the distribution of drugs for one ADR.
4. In Figure 7, nodes are colored in different colors. This means different clusters as described in the text. It is not clear how to cluster nodes.
5. For seven classification algorithms, some used default parameters, whereas the parameters of some algorithms were tuned. Why?
6. In Table 3, the models were constructed by adding network features one by one. However, how to determine the adding sequence? Why degree centrality was added first, followed by weighted degree centrality, eigenvector centrality, etc.?
7. Line 339-340, “Some other notable figures include pyelonephritis” seems not correct. What is figure?
8. To prove the importance of network features, the models only using them should be constructed and evaluated. The comparisons between these models and those only using drug features are necessary to be added.
9. Based on the performance of models in Liu’s study (reference 11 in this study), the models in this study provided lower performance. Please further give the value of this study.
10. The evaluation of models was not very rigorous. Authors constructed the drug-to-drug network first and extracted ten network features for each drug. This means that the network features for one drug do not change. When evaluating the performance of one model for one ADR using ten-fold cross-validation, the ADR information of test drugs should be removed from the drug-to-drug network and then the network features can be extracted from such network.
Reviewer 2 Report
In this paper, Authors tried to construct a drug-to-drug network based on non-clinical data 14 sources. Network presents underlying relationships between drug pairs according to their com-15 mon ADRs. Then multiple node-level and graph-level network features are extracted from this net-16 work, e.g. weighted degree centrality, weighted PageRanks etc. They fed into seven machine learning models, e.g. Lo-18 gistic Regression, Random Forest, Support Vector Machine, etc., and were compared to the baseline 19 where there were no network-based features considered. These experiments conclude that all the 20 tested machine-learning methods would benefit from adding those network features. Among all 21 these models, Logistics Regression (LR) shows the highest mean AUROC score (82.1%) across all 22 ADRs tested. In this case, the paper has an interesting results and recommend to publish i in Healthcare Journal.
Author Response
Thank you very much for your comments!
Reviewer 3 Report
The authors raised a very important topic related to the safety of drugs.It is worth completing the ADRs classification and division in the introduction.
I would be very pleased and interested to read sample analyzes added as a supplement.
Round 2
Reviewer 1 Report
My comments have been addressed. The current version can be accepted.